# Disruptions in gene interaction networks abolish host susceptibility to *Trichostrongylus colubriformis* infections in sheep

Fang Liu[1,2]*, Jody McNally[3], Jonathan Shao[2], Aaron B. Ingham[4], Peter W. Hunt[3]*, Robert W. Li[2]*

**1** College of Public Health, Zhengzhou University, Zhengzhou, China, **2** USDA-ARS, Animal Parasitic Diseases Laboratory, Beltsville, Maryland, United States of America, **3** CSIRO F.D. McMaster Laboratory, Armidale, Australia, **4** CSIRO Agriculture and Food, St. Lucia, Australia

* fliu19910205@gmail.com (FL); Peter.Hunt@csiro.au (PWH); Robert.li@usda.gov (RWL)

## Abstract

The roundworm *Trichostrongylus colubriformis* is one of the most important parasites in sheep, impairing feed conversion and reducing growth. However, the molecular mechanism of host resistance to this important species remains elusive. In this study, we compared gene interaction networks manifested in the jejunum transcriptome between sheep bred for parasite resistance (RES) and susceptibility (SUS) in response to a *T. colubriformis* challenge infection. Selections for host resistance compromised parasite establishment and resulted in a 98.8% reduction in worm burden ($P < 0.00001$; $N = 20$ per group). 319 genes displayed a significant difference in transcript abundance between RES and SUS phenotypes. The gene encoding lectin, galactoside-binding, soluble, 15 (*LGALS15*) had significantly higher expression in RES than SUS lambs. Enriched pathways included antigen processing and presentation and Fc gamma R-mediated phagocytosis. Two algorithms, weighted gene co-expression network analysis (WGCNA) and differential gene correlation analysis (DGCA), were applied to infer gene interaction networks. Both algorithms demonstrate that genes in SUS lambs appear to be more closely correlated than in RES lambs. WGCNA identified a module that was positively correlated with worm counts only in SUS animals ($R = 0.67$; $P = 0.001$). DGCA detected approximately four times more unique gene correlation pairs in SUS than in RES lambs. Gene Ontology analysis among the genes with gain-of-correlations shows marked differences in cell division, such as mitotic cytokinesis, sister chromatid segregation, and kinetochore, between the two phenotypes. Correlated genes unique to RES can be used for the development of alternative biomarkers to facilitate breeding. Moreover, dietary approaches to disrupt close gene interactions within key modules may have the potential to reduce worm burden. It is conceivable that feeds, phytochemicals or feed additives that promote specific gene interactions can be used for sustainable parasite controls.

**Data availability statement:** The datasets generated and/or analyzed during the current study are available in the NCBI SRA repository (SRA accession# PRJNA1214723). Raw counts and FPKM values for all genes can be freely accessed at https://data.mendeley.com/datasets/djjkp2743h/1. All other supporting materials can be found in the online Supporting Information.

**Funding:** This study is partially supported by USDA-ARS (Grant # 58-8042-3-022-F to PWH) and by Henan Provincial Science and Technology Research Projects (Grant# 242102311161 to FL). The funders had no role in study design, data collection and analysis, decision to publish, or preparation of the manuscript.

**Competing interests:** The authors have declared that no competing interests exist.

## Author summary

The worms from the genus *Trichostrongylus* are ubiquitous in distribution and are of high relevance to food animal production and human health due to their zoonotic potential. Rapid emergence and spread of anthelmintic resistance genes in farm animals have markedly compromised the efficacy of existing anthelmintic drugs, a mainstay of parasite control strategies in food animal production. Breeding for resistance to worms represents a cost-effective strategy to mitigate parasitic diseases. However, molecular mechanisms underlying resistance remain largely unknown. Here we took advantage of unique sheep populations bred for resistance and susceptibility. We identified 319 genes with significant differences in abundance between the two phenotypes as well as many correlated gene pairs unique to resistance or susceptibility. We uncovered multiple clusters of genes that were significantly correlated with parasitological traits. Our findings provided insights into the regulatory control of host resistance. Sheep farmers will benefit from this study by having biomarkers indicative of resistance for selective breeding.

## Introduction

Gastrointestinal (GI) nematode infections in small ruminants remain a major impediment to efficient food animal production. Parasitism affects productivity, including reducing growth and milk yield in sheep [1]. In young animals, particularly during their first grazing season, parasite infection is also a leading cause of mortality [2]. Potential economic loss due to GI nematode infections is widely recognized by farmers and veterinarians.

The geographical distribution of GI nematodes in host species is heavily influenced by climate (such as rainfall) and environmental factors, such as temperature and humidity. Species from the order *Strongylida*, including *Haemonchus contortus, Teladorsagia circumcincta,* and *Trichostrongylus* species, are very common in small ruminants. *Trichostrongylus colubriformis* is a ubiquitously distributed parasite in small ruminants, most prevalent in North America, East and South Africa, and Australia [3,4]. For example, the prevalence of *Trichostrongylus* species in U.S. goat operations can be as high as 97.2%, five times higher than that of *Haemonchus* in the same operations, according to the 2019 USDA National Animal Health Monitoring System Goat Study.

Moreover, *Trichostrongylus* species are of high relevance to human health. Since the first reported human case in USA in 1938, dozens of publications have described human trichostrongylosis in at least 29 countries around the globe, from Australia to Iran, and from Brazil and Laos to Italy [5]. While low in general populations in developed countries, the prevalence of *Trichostrongylus* infections in humans in developing countries, particularly in the communities with relatively poor hygiene and in close proximity with livestock, is high. For example, a recent study from Guilan, Iran, a

subtropical region with heavy rainfalls, shows a high prevalence of *T. colubriformis* infection, up to 34.4% among villagers [6]. In a tropical region of Laos, where intimate human and livestock interactions are common, 36.9% of villages examined had signs of *Trichostrongylus* infection [7]. At least 11 of the approximately 30 species described in the genus *Trichostrongylus*, such as *T. orientalis*, *T. colubriformis*, *and T. axei*, are zoonotic and can infect humans [8–10].

The clinical sign of human trichostrongylosis is generally asymptomatic to mild, depending on worm burden. The symptoms are often complicated by co-infections with worms from other genera. People with heavy worm burden display digestive disorders, including diarrhea, abdominal pain and loose stool [7]. Weight loss, rash, and anemia associated with hypereosinophilia are also observed [8,11]. In livestock species, challenge infection with 2,500 *T. colubriformis* infective larvae (L3) for 13 weeks in growing lambs results in severe pathology, including small intestine lesions that lead to impaired digestion and absorption. As a consequence, the infected lambs had a 37% reduction in daily weight gains and significantly lowered feed conversion, compared with uninfected controls [12,13].

Direct interactions between human epithelial cells and *T. colubriformis* larvae have been demonstrated using an in vitro air–liquid interface co-culture system. *T. colubriformis* motion at the infection site can induce intestinal epithelial cell necrosis [14]. *T. colubriformis* larvae and their excretory/secretory products strongly stimulate the alarmin IL33 expression in human epithelial cells. The alarmin released by non-immune cells plays a critical role in the initiation of Th2 immune responses that drive parasite expulsion and ameliorate host tissue injury [15]. *In vivo*, infections by *T. colubriformis* elicit a strong host immune response, compared with uninfected controls [12]. The infected lambs have an increased number of inflammatory cells in the mucosa and an increased production of *T. colubriformis*-specific antibodies (particularly IgG and IgA) and eosinophilia [12] and mucus secretion [16]. An earlier microarray study identified a classical Th2 type immune response to *T. colubriformis* challenge infection in draining lymph nodes, which contain approximately 15% dendritic cells and 85% lymphocytes, with upregulation of IL13 [17]. A global downregulation of many genes involved in immune functions, including antigen presentation, caveolar-mediated endocytosis, and protein ubiquitination in response to the infection, was also evident. Other studies have demonstrated there exist large variations in host immune responses to both natural and challenge infections between breeds [18,19] and among individual animals within the same breed [13,18,20].

While a robust immune response is often associated with resistance to infection, manifestations of parasitism in sheep, such as reduced appetite, weight loss, and diarrhoea, are due to immune-mediated pathology rather than direct effects of the parasite [21]. Expressed resistance to GI nematode infections in ruminants includes decreased worm establishment, arrested development of invading larvae, reduced parasite fecundity, and enhanced worm expulsion. While the parameters of resistance are not well defined, a substantial reduction in both fecal worm egg counts (WEC) and worm burden in resistant animals is expected. Several previous studies have demonstrated that selection for the divergence in WEC can be achieved. In Romney sheep, the mean WEC in control lambs was 1,255 eggs/g of feces, whereas WEC for the high and low selection lines were three and 0.27 times than the control mean (WEC of 3853 and 339 eggs/g, respectively) [22]. In Merino sheep, over three decades of selective breeding have generated resistant and susceptible lines with well differentiated and stable phenotypes [23]. Lambs from resistant lines released 12.8 times fewer eggs than susceptible lines grazing on the same pasture in our study (WEC = 240.00 ± 209.57 and 3078.00 ± 995.01, respectively; *P* = 1.12 x $10^{14}$, *N* = 237). It is known that there exist substantial transcriptome alterations and disruptions underlying pathophysiological events during parasitic infection, especially at the site of infection [24]. Earlier studies show that resistant sheep breeds are able to more rapidly up-regulate Th2 cytokines than susceptible sheep breeds [25]. Host tissue transcriptome characteristics in resistant lambs or resistant sheep breeds to challenge infections by *H. contortus* and *T. circumcincta* have been subsequently examined using RNAseq technologies [18,26]. However, locally-adapted breeds, such as Santa Ines, Suffolk and Ile de France, did not show any significant differences in worm burden and WEC and had similar cellular and immune responses to *T. colubriformis* [20], which poses special challenges to understand the mechanistic bases underlying host resistance to this important parasite species in these breeds. Moreover, genes and pathways at the site of infection in lambs selected for resistance and susceptibility in response to *T. colubriformis* infections are largely unknown.

Moreover, gene interaction patterns underlying the resistance phenotype within a breed have yet to be elucidated. In this study, we characterized gene interaction networks in the jejunal transcriptome in our unique sheep populations that have undergone decades' selective breeding for resistance and susceptibility using RNAseq and advanced network algorithms.

## Materials and methods

### Ethics statement

All animal handling procedures were carefully reviewed, approved, and monitored by the the Commonwealth Scientific and Industrial Research Organisation (CSIRO) Armidale Animal Ethics Committee (Animal Research Authority Numbers# 10/22 and 08/23). The animal use protocol and procedures were in compliance with the Australian Code for the Care and Use of Animals for Scientific Purposes and all relevant Commonwealth, State and Territory legislation on animal welfare.

**Animal studies.** The animal study was conducted at the Commonwealth Scientific and Industrial Research Organisation (CSIRO) F.D. McMaster Laboratory in Armidale, Australia, as previously described [27]. Fecal samples from the sheep in the CSIRO parasite selection lines were collected. These animals grazed on the same pasture during their first grazing season. WEC data were obtained using the McMaster method. The animal study was conducted to compare the difference between the parasite resistance (RES) and susceptibility (SUS) lines from each of the two selection flocks, *Haemonchus* selection (HSF) and *Trichostrongylus* selection flocks (TSF). The WEC values obtained from the lambs on pastures were used to rank and select the animals for subsequent studies. On the 2nd day after their transfer from the pasture to animal house pens, all animals were drenched using a mixture of commercial dewormers, Startect, Zolvix Plus, Flukazole, Rametin, and Closantel, according to their recommended doses, to remove existing parasites, including nematodes, fluke and tapeworms. The animals were also vaccinated using Websters 6-in-1 at a dose of 1 mL per head, to protect against clostridial diseases and *Corynebacterium* infections. These animals were housed in four pens with water troughs and environmental enrichment. The animals were fed 700 g/day of animal house pellets (lucerne-based pellets) plus heat-treated chaff and had free access to water. On the 13th day of 14 days' acclimation, feces were sampled from each animal and analyzed to detect any roundworm eggs. No lambs with WEC values greater than zero were observed. Another exclusion parameter was animal behavior and feeding habits. Based on these criteria, 40 wether lambs (20 resistant and 20 susceptible animals), aged 10–15 months, were weighed and identified for a subsequent challenge infection experiment (see S1 Table for sample metadata). On the 14th days after the transfer, each animal received 20,000 *T. colubriformis* infective larvae in a single dose administered orally. The infection was allowed to progress for 14 days. On the day before the necropsy, the animals were weighed and not fed. All lambs were euthanized by the captive bolt stunning method followed by exsanguination. Jejunum tissue samples were collected at necropsy and snap-frozen in liquid nitrogen for total RNA extraction. Entire jejunum contents and mucosal scrapings were then collected. The intestinal contents and scrapings were combined and brought to 500 ml total volume using running water. The jejunum samples were mixed well; and two 30-ml subsamples were then taken, frozen at -20ºC, and later thawed and worms were counted under a dissection microscope. The worms in the 2 x 30-ml subsamples represented 12% (60/500) of the total sample collected. Total worms, including males, females, and immature larvae, were recorded.

**RNA extraction, RNAseq library preparation and sequencing, and quantitative RT-PCR.** Total RNA was extracted from the jejunum tissue using TRIzol reagents, followed by DNase digestion and purification using a Qiagen RNeasy Micro Kit (Qiagen, Germantown, MD, USA). RNA integrity was verified using a BioAnalyzer 2100 (Agilent, Palo Alto, CA, USA). Quantitative reverse transcription (RT) PCR reactions were carried out as previously described [28] and the primers used can be found in S2 Table. The amplification reactions were subjected to an initial denaturation at 95 °C for 3 min, followed by 40 cycles of 95 °C for 30 s, 60 °C for 30 s, and 72 °C for 30 s. A standard-curve based absolute quantification method was used.

RNAseq libraries were prepared using a Qiagen QIAseq Stranded RNA Library Enzyme kit and a Unique Dual Index (UDI) kit. The depletion of rRNA molecules was achieved using a QIAseq FastSelect -rRNA HMR kit, which enables the

removal of sheep rRNA species up to 95.4% according to the technical specification by the manufacturer. Paired-end sequences were generated at 150 bp/read using an Illumina Novaseq 6000 sequencer. Raw sequence reads have been deposited to the NCBI Sequence Read Archive (SRA) database (Accession# PRJNA1078452). The raw counts and fragments per kilobase of transcript per million fragments mapped (FPKM) values are also freely available (https://doi.org/10.17632/djjkp2743h.1 and https://data.demo-uni.com/datasets/djjkp2743h/1).

**Bioinformatics and network inferences.** The mean number of raw reads is 60,821,935 ± 15,225,107.28 per sample with the Phred quality score Q20 = 97.44 ± 0.27 (N = 40). The basic statistics of these raw reads are available in S3 Table. Raw sequences were trimmed using Trimmomatic with default parameters (v0.38). The preprocessed reads were analyzed using the Hisat2-String Tie-DESeq2 pipeline (v1.48.1) with default parameters [29]. The sheep genome assembly ARS-UI_Ramb_v3.0 (NCBI RefSeq assembly GCF_016772045.2) was used as the reference genome. Gene enrichment analysis was conducted using the DAVID pipeline (v2023q4) [30]. The raw count data were converted to FPKM values as the data matrix for subsequent network inference.

Weighted correlation network analysis (WGCNA) was used to construct gene interaction networks based on the FPKM data matrix (v1.70.3) [31]. The *goodSamplesGenes* function was applied to filter genes with low variance. The signed network was inferred based on a biweight midcorrelation (bicor) method. The soft threshold power was set to 0.85. The minimum module size was 30. The module preservation was calculated using the *modulePreservation* function in the package. Both composite preservation statistics, *Zsummary* and *medianRank*, were derived. Empirical thresholds, Zsummary < 10 or medianRank > 8, were used as the cutoff for non-preserved modules. Hub genes were defined as those with absolute module membership ($k_{ME}$) values ≥ 0.95.

Differential correlation analysis was performed using the Differential Gene Correlation Analysis (DGCA, v1.0.0) R package [32]. For the global analysis, the genes were first filtered based on median FPKM values. The resultant input list included 15,000 genes with annotations. Correlation analysis of gene pairs was conducted using the Spearman rank method in the package. The permutation-based hypothesis tests were used to rank differentially correlated gene pairs between the RES and SUS groups (FDR or q value < 0.01). Gene pairs were further classified into gain-of-correlation or GoC (Spearman's rank correlation coefficient rho $\rho \geq 0.9$) or loss-of-correlation categories (LoC; $\rho \leq 0.3$). Gene Ontology (GO) enrichment of differentially correlated gene pairs with GoC in RES and SUS groups were further analyzed using the GOstats R package v2.74.0 (adjusted $P < 0.01$ as significance cutoff). The module detection was carried out using the Multiscale Embedded Gene Co-expression Network Analysis (MEGENA, v1.4.1) algorithm [33] based on the differential correlation output from DGCA.

**Statistical analysis.** The R Stats package (R version 4.3.0) was used for statistical analyses. A non-parametric Wilcoxon Rank Sum test was used for two-group statistical comparisons. The number was expressed as means ± SD, unless stated otherwise. False discovery rate (FDR) corrections were also applied to unadjusted $P$ values, where a significance threshold was considered at FDR-adjusted $q$ values ≤ 0.05.

## Results

### Lambs bred for parasite resistance developed unique transcriptome characteristics in response to T. colubriformis challenge infection

Worm establishment was significantly impaired in resistant lambs. Compared to susceptible lambs, resistant lambs had a 98.8% reduction in worm burden (two-tailed Wilcoxon $P < 0.00001$; N = 20 per group; Fig 1A). The worm count data from the challenge infection was strongly correlated with the WEC value obtained when lambs were grazing on pasture (Spearman rank rho = 0.828).

Underlying parasitological events during infection are transcriptome alterations, especially at the site of infection. Overall transcriptome characteristics of resistant and susceptible lambs in response to *T. colubriformis* challenge infection

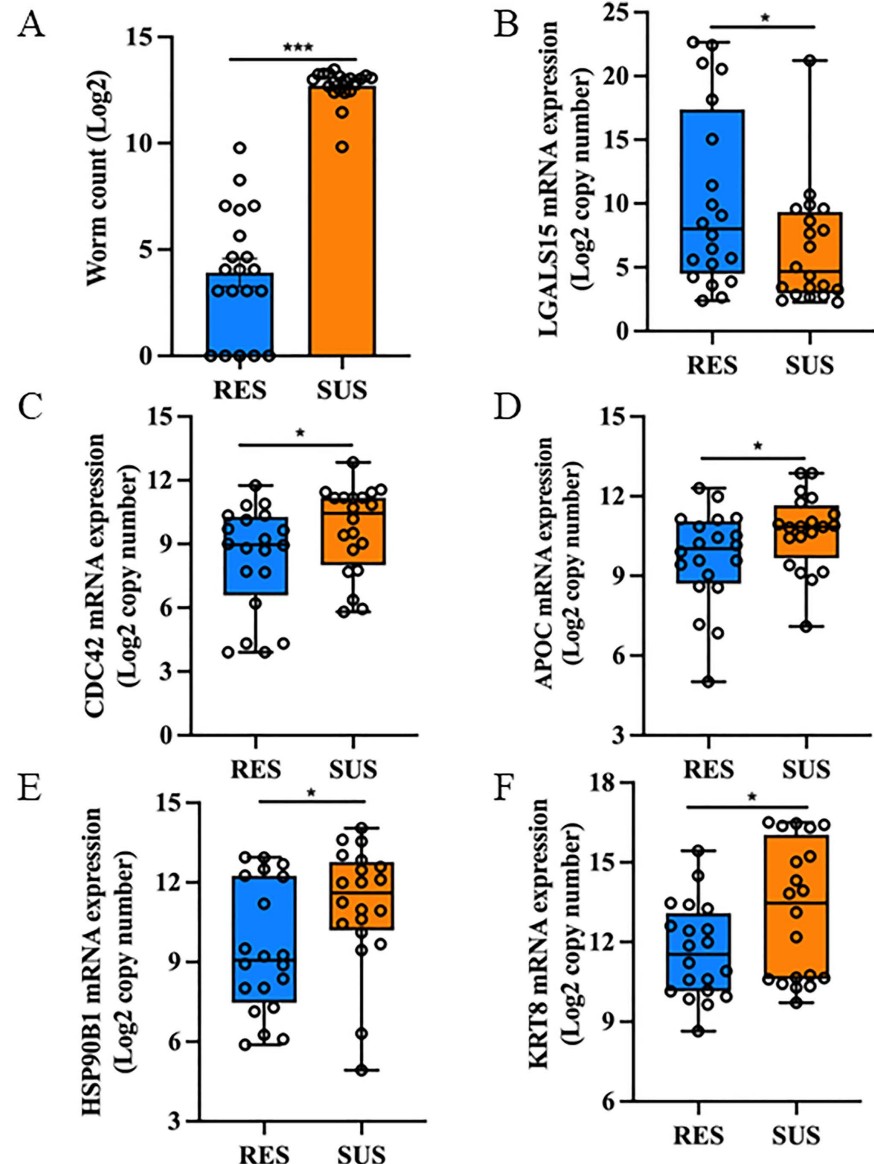

**Fig 1. Worm burden and expression of select genes at mRNA levels detected using quantitative RT-PCR in resistant (RES) and susceptible (SUS) lambs in response to a *Trichostrongylus colubriformis* challenge infection in sheep.** A. parasite worm counts. B – F: the expression of select transcripts detected using quantitative RT-PCR. Y-axis: transcript copy number expressed as $\log_2$ values. B. *LGASL15*. C. *CDC42*. D. *APOC*. E. *HSP90B1*. F. *KRT8*. N = 20 per group. * $P < 0.05$; ** $P < 0.01$; *** $P < 0.001$ (based on the Wilcoxon rank sum test).

were similar, as judged by the indistinguishable phenotype groups in the principal component analysis plot (PCA; S1 Fig). The first component explained 90.3% of the total variance while the second component explained an additional 5.7% of the variance, suggesting that PCA plots described the overall transcriptome structure rather well. Nevertheless, the jejunal transcriptome of resistant lambs differed markedly from that of the susceptible lambs in response to a 14-day challenge infection. A total of 319 differentially expressed genes (DEG) were detected using a combined cutoff value of adjusted *P* values (FDR < 0.05) and absolute fold changes ≥ 1.5 (S4 Table). The majority of these DEG had a significantly higher abundance in susceptible than resistant lambs. Among the seven DEG with higher transcript abundance in the

resistant lambs, five of them belongs to the variable chain T cell receptor genes (TR-V gene). Lectin, galactoside-binding, soluble, 15 (*LGALS15* or galectin 11) had significantly higher expression in resistant lambs (5.1 fold); its expression was subsequently confirmed using quantitative RT-PCR (Fig 1B). Nevertheless, the other two galectins detected, *LGALS3* and *LGALS4*, were significantly more abundant in susceptible lambs. At least seven actin-related genes, including *ACTB*, *ACTG*, *ACTR2*, *ACTR3*, *ARPC2*, *ARPC3*, and villin 1 (*VIL1*) were also more abundant in susceptible than resistant lambs. Similarly, five apolipoproteins, including A1, A4, B, C3, and apolipoprotein B mRNA editing enzyme catalytic subunit 3A (*APOBEC3A*) and two fatty acid binding proteins (*FABP1* and *FABP2*), were also significantly more abundant in the jejunal transcriptome of susceptible lambs. The expression of C-C motif chemokine ligand 25 (*CCL25*), *CD74*, and cell division cycle 42 (*CDC42*) was also higher in SUS than RES lambs in response to challenge infection. The expression of *CDC42*, *APOC*, heat shock protein 90 beta family member 1 (*HSP90B1*), and keratin 8 (*KRT8*) was also confirmed using quantitative RT-PCR (Fig 1C–1F).

The DEG with significantly higher transcript abundance in susceptible lambs were further annotated against the UniProt Knowledgebase (UniProtKB) to identify the enriched protein function (S5 Table). The proteins encoded by these DEG were involved in a broad range of biological processes, such as lipid metabolism, transport, and tricarboxylic acid (TCA) cycle (FDR = 0.0175) and exerted some basic molecular functions, including actin-binding (FDR = 0.0183), monooxygenase (FDR = $6.84 \times 10^{-5}$), and oxidoreductase activities (FDR = $5.36 \times 10^{-7}$). Of note, the proteins encoded by these DEG were also involved in post-translational modifications (PTM; S5 Table). For example, five PTM processes were significantly enriched, including acetylation (FDR = $8.84 \times 10^{-10}$), isopeptide bond (FDR = 0.0145), methylation (FDR = $1.63 \times 10^{-3}$), phosphoprotein (FDR = 0.0145), and ubiquitin-like protein (Ubl) conjugation (FDR = 0.0487).

These DEG were also annotated using the DAVID pipeline [34]. At least 34 Kyoto Encyclopedia of Genes and Genomes (KEGG) pathways were significantly enriched among these DEG at FDR < 0.05 (S6 Table). These pathways were involved in a wide array of cellular functions, from metabolism, signaling, to secretion. For example, the proteins encoded by the DEG were involved in ascorbate and aldarate metabolism (FDR = 0.0315), carbon metabolism (FDR = 0.0102), drug metabolism (cytochrome P450 and other enzymes), fatty acid metabolism (FDR = 0.0419), and retinol metabolism (FDR = 0.0030). Other enriched KEGG pathways included peroxisome proliferator-activated receptor (PPAR) signaling and retinol metabolism (FDR = $2.16 \times 10^{-4}$ and $30.4 \times 10^{-3}$, respectively). Both PPAR and retinoic acid signaling pathways play an important role in modifying host-parasite interactions [35,36]. Furthermore, regulation of actin cytoskeleton and tight junction were also enriched in these DEG.

## Gene interaction networks and modules associated with parasitological traits

Host resistance to parasitic infection is a complex trait that results from concerted efforts of hundreds of genes located in numerous genomic loci. Genes do not function in isolation but interact with each other to participate in various molecular processes and carry out biological functions. The expression of genes is also tightly regulated by environmental cues, regulatory elements such as promoters, enhancers, and silencers, transcriptions factors and cofactors, and epigenetic landscapes, leading to the formation of spatiotemporal interaction networks. Gene correlation network analysis allows us to have a holistic view of cellular processes and gain novel insights into the mechanisms underlying the development and manifestation of host resistance. Towards this end, we performed gene interaction network analysis in RES animals using the WGCNA algorithm based on the biweight mid-correlation method. Three modules (ME), or clusters of highly interconnected genes, were detected in the RES network and are referred to blue, turquoise and grey (S2 Fig). Module blue (MEblue) had 77 members (nodes), including *CD2*, *CD81*, interleukin 2 receptor subunit gamma (*IL2RG*), septin 6 (*SEPTIN6*), and several actin-related and cytoskeleton-associated genes (*ARPC5*, *CKAP5*, and *MACF1*). Several genes, including arginine and serine rich coiled-coil 2 (*RSRC2*), pyruvate kinase M1/2 (*PKM*), RAB11 binding and LisH domain, coiled-coil and HEAT repeat containing (*RELCH*), and vacuolar protein sorting 13 homolog B (*VPS13B*), acted as hub genes, as defined by high module membership values ($k_{ME} > 0.90$) and absolute Gene Significance measures ($GS > 0.5$)

in this module. Intriguingly, module eigengene values of MEblue were negatively correlated with the parasitological trait WEC (R = − 0.53; P = 0.02; S2 Fig), suggesting that WEC measures decreased with increasing expression of the genes in this module, as represented by its eigengene value. Module turquoise (MEturquoise) is also interesting. This module contained 2,836 genes and was also negatively (but marginally) correlated with WEC (R = − 0.41; P = 0.08; S2 Fig). Polymeric immunoglobulin receptor (*PIGR*), calnexin (*CANX*), ATP synthase F0 subunit 6 (*ATP6*), FAT atypical cadherin 1 (*FAT1*), heat shock protein 90 beta family member 1 (*HSP90B1*, Fig 1E), and ATPase plasma membrane Ca2 + transporting 1 (*ATP2B1*) were among the hub genes in this module (all with module memberships > 0.99). Gene enrichment analysis suggests that at least four major types of cell junctions, including adherens, focal adhesion, gap junction, and tight junctions, were significantly enriched in this module at FDR < 0.05 (S7 Table). Furthermore, multiple pathways associated with host immune responses, such as antigen processing and presentation, B cell receptor signaling pathway, chemokine signaling pathway, C-type lectin receptor signaling pathway, Fc gamma R-mediated phagocytosis, leukocyte transendothelial migration, platelet activation, and T cell receptor signaling pathway, were also significantly enriched in MEturquoise. Not coincidentally, 93.1% of all 319 DEG identified also belonged to this module. All DEG in this module were significantly more abundant in susceptible lambs.

Unlike the RES network, the gene interaction network in the susceptible lambs (SUS network) were clustered in 159 modules based on the same cutoff threshold. Module lightcyan1 had a moderate yet positive correlation with worm counts (R = 0.67; P = 0.001; Fig 2), whereas both Modules pink3 and tan3 were marginally correlated with all parasitological traits measured, WEC (both natural and $\log_2$ transformed) and worm counts (P < 0.05). Module lightslateblue was strongly and positively correlated with the selection flocks (HSF vs TSF; R = 0.83; P = 5.0 x $10^{-6}$). Further examination of module compositions in the SUS network uncovered that the percentage of non-coding RNA in MElightcyan1 was disproportionately high; and 48% of the 96 module members were not protein-coding, including 29 long non-coding RNA (lncRNA) genes, seven small nuclear RNA (snRNA) genes, five small nucleolar RNAs (snoRNAs) genes, and one microRNA gene (oar-mir-221) as well as two TR-V genes.

We also investigated the properties and reproducibility of modules in gene interaction networks and their change across two phenotypes RES and SUS, using the *modulePreservation* function in the WGCNA package. MEblue in the RES and SUS networks contained 77 and 4,700 members, respectively. 68 of the 77 members in this module in the RES network were also present in the blue module in the SUS network (Fig 3). The RES and SUS blue modules were similar as judged by both composite statistics: $Z_{summary}$ and medianRank. Generally, $Z_{summary}$ values, which measure both density and preservation of connectivity, between 2 and 10, suggest weak to moderate evidence in support of module preservation, while $Z_{summary}$ >10 presents strong evidence for preservation. MEblue had a $Z_{summary}$ value >45, suggesting this module is highly preserved between RES and SUS networks. Moreover, the status of all top five hub genes in the module was preserved between the two networks. The gene enrichment analysis showed that the UniProt KW biological process related to glycolysis was enriched in MEblue. However, we found limited evidence that many of the other modules in the SUS network were preserved in the RES network, particularly those modules positively correlated with WEC and worm counts, suggesting that decades' selective breeding disrupted the network substructure and global gene expression patterns that promoted host susceptibility to *T. colubriformis* infection.

### Functional implications of gene correlation networks in the susceptible lambs

A global gene correlation pattern between the RES and SUS phenotypes was also compared using Spearman rank-based analysis in the DGCA package. After filtering out the genes with lower median FPKM expression values, the top 15,000 genes with annotations were used as input. A total of 255,753 significantly differentially correlated gene pairs were identified at a cutoff threshold FDR < 0.05 (see Mendeley data, https://doi.org/10.17632/tkp7kh994w.1). Among them, 6,704 positive correlations in both RES and SUS groups (the class +/+) and 440 negative correlations in both (the class -/-) gene pairs were detected. For example, heat shock protein 90 alpha family class A member 1 (*HSP90AA1*) was strongly

**Fig 2. Select modules significantly correlated with parasitological traits in the gene interaction network inferred from susceptible lambs in response to *T. colubriformis* infection.** The color bar on the left represents module names. The numbers denote correlation coefficient values obtained using the biweight mid-correlation (bicor) method (Right). The number in the parenthesis represents the significance (*P* value). WEC: fecal worm egg counts. $N = 20$ per group.

and positively correlated with KDEL endoplasmic reticulum protein retention receptor 2 (*KDELR2*) in both RES and SUS groups ($\rho = 0.99$, FDR = 1.00 x 10$^{-16}$ and $\rho = 0.86$, FDR = 1.52 x 10$^{-6}$, respectively). *PIGR* was positively correlated with 22 genes regardless of the RES and SUS groups. For example, *PIGR* was strongly and positively correlated with ribosomal protein L13a (*RPL13A*) and serpin family B member 1 (*SERPINB1*) in both RES and SUS. On the other hand, *PIGR* was also negatively correlated with both homeobox B5 (*HOXB5*) and potassium voltage-gated channel subfamily A member 1 (*KCNA1*) in both RES and SUS phenotypes. There were 22,684 gene pairs where a positive correlation was observed in RES lambs but not in SUS lambs (class + /0). For example, *CD164* and poly(rC) binding protein 1 (*PCBP1*) were positively correlated only in RES ($\rho = 0.95$, FDR = 2.43 x 10$^{-10}$) but not in SUS (Fig 4). Four to five times higher numbers of gene pairs were correlated in SUS lambs, but not in RES lambs, compared to those correlated in the RES but not SUS sets

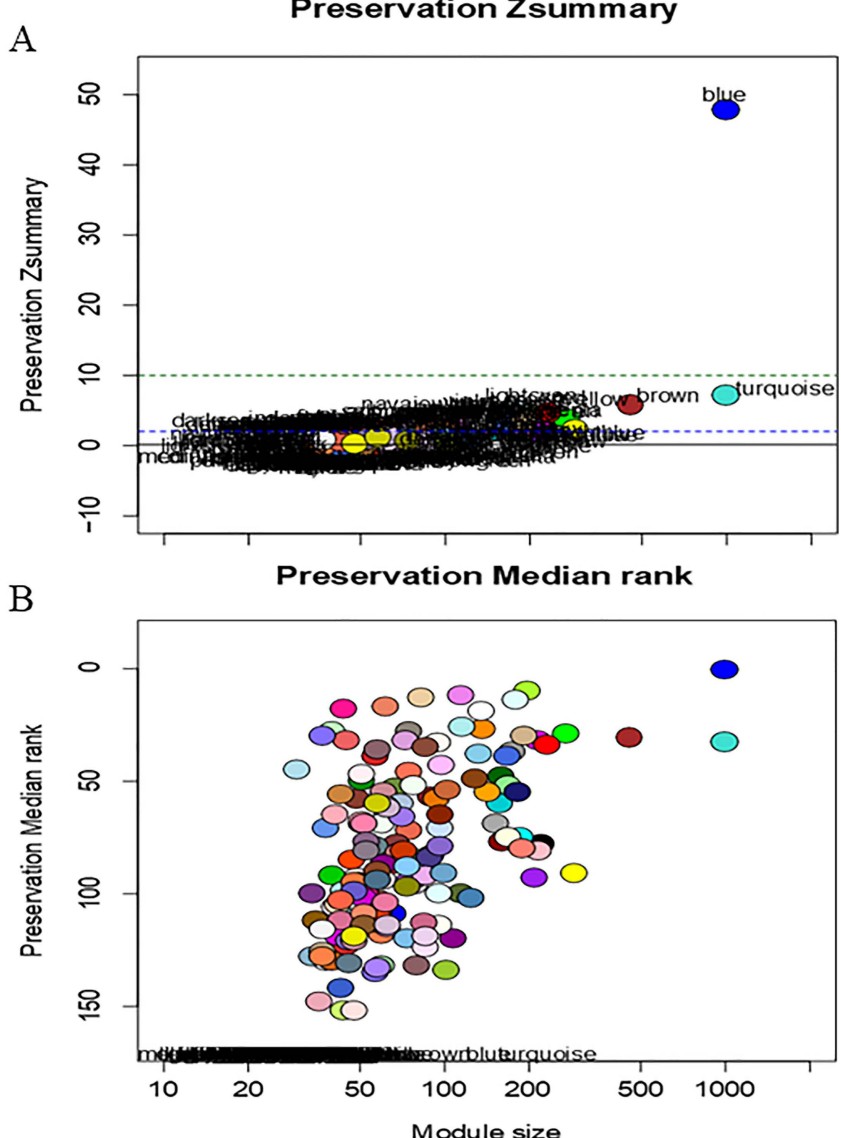

**Fig 3. Composite statistics of modules inferred from resistant (RES) lambs that were preserved in susceptible (SUS) lambs.** A. the summary statistic $Z_{summary}$ (y-axis) as a function of the module size (x-axis). Each dot represents a module as labeled by a color. The dashed lines indicate empirical thresholds for $Z_{summary}$ values. Any module above the upper dash line with $Z_{summary} > 10$ would have strong evidence that the module is preserved; The blue module in the RES network, in this case, was strongly preserved in the SUS network. The $Z_{summary}$ range between 2 and 10, such as in Module turquoise, indicates that there is weak to moderate evidence of module preservation, whereas $Z_{summary} < 2$ suggests that little evidence for module preservation.

(89,322 GOC and 102,539 LoC gene pairs). A strong and negative correlation between leucine rich repeat containing G protein-coupled receptor 4 (*LGR4*) and cysteine and serine rich nuclear protein 3 (*CSRNP3*) was only evident in SUS (S3 Fig; $\rho = -0.98$, FDR = $1.82 \times 10^{-14}$). In addition to the examples above, *PIGR* was also found to be significantly correlated with other genes in either the RES or SUS groups, rather than both. For those genes positively correlated in one phenotype but not in another phenotype, seven times more genes were positively correlated with *PIGR* in the SUS group (21 genes) than in the RES group (3 genes).

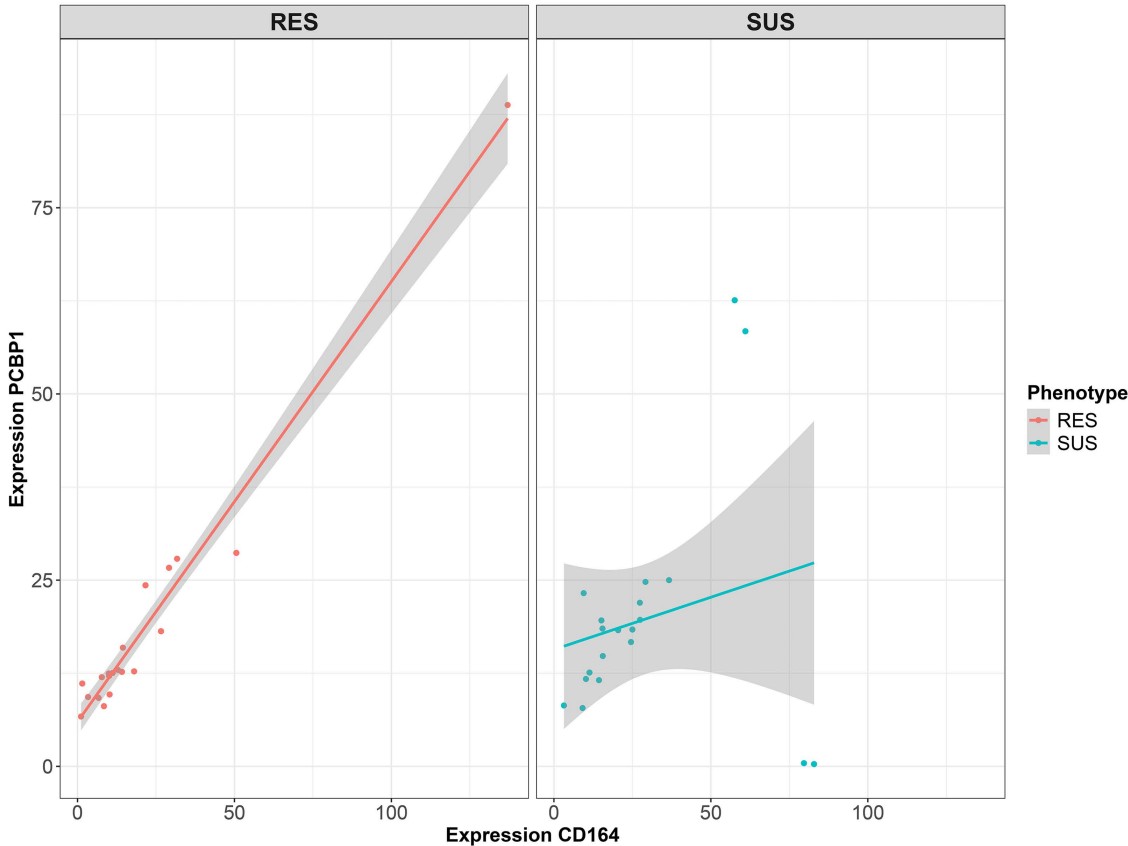

**Fig 4. Differentially correlated gene pairs between the SUS and RES groups calculated using Spearman's rank correlation.** Expression values between *CD164* (x-axis) and poly(RC) binding protein 1 (*PCBP1*; y-axis) in the resistant phenotype (RES, left) and the susceptible lambs (SUS, right) were shown. The grey area represents 95% confidence intervals. The red (left) and blue (right) lines represent a linear model of the best fit. The correlations between *CD164* and *PCBP1* were strong in resistant (RES) lambs (Spearman's rho $\rho = 0.94$, *FDR* = 2.43 x 10$^{-10}$), but no correlation in susceptible (SUS) lambs ($\rho = 0.31$, *FDR* = 0.18).

A global view of the enriched GO terms among genes with GoC displayed a similar pattern across the RES and SUS groups (Fig 5). Three GO biological process (BP) terms associated with RNA splicing, including mRNA splicing via spliceosome or via transesterification reactions showed an identical odds ratio (OR) in both phenotypes (OR = 2.8). RNA (and mRNA) binding was also enriched in both phenotypes with similar OR values. However, several cell division-related GO terms belonging to BP and cellular components (CC), such as mitotic cytokinesis, sister chromatid segregation, centromeric regions, and kinetochore, had higher OR in susceptible than resistant lambs (Fig 5).

Further analysis of the genes with GoC that are unique to each phenotype, i.e., those GoC genes in the classes + /0 and 0/+ in respective RES and SUS groups, shows a different story (S8 Table). Among genes with GoC in RES only, two Molecular Function (MF) terms, actin filament binding and protein binding were significantly enriched (FDR = 0.0044 and 7.65 x 10$^{-7}$, respectively). However, among the genes with GoC unique to SUS, enriched GO MF terms were much broader, including the above mentioned two terms but also including 10 additional MF terms, such as ATP binding, DNA-binding transcription activator activity, RNA polymerase II-specific, DNA-binding transcription factor activity, RNA polymerase II-specific, GDP binding, identical protein binding, mRNA binding, protein domain specific binding, protein homodimerization activity, sequence-specific DNA binding, and ubiquitin protein ligase binding.

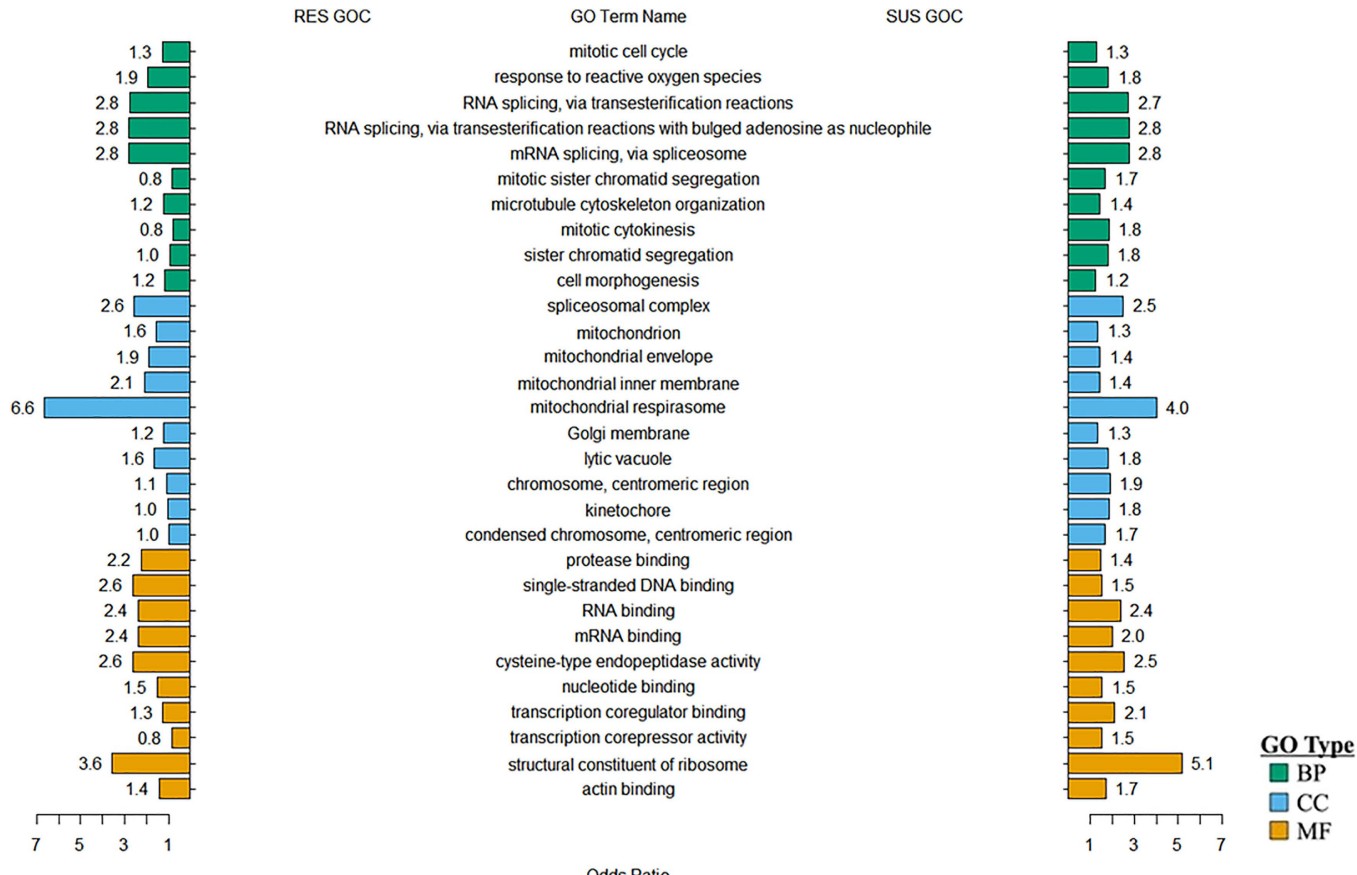

**Fig 5. A global view of gene ontology (GO) terms enriched in the set of gene pairs with gain-of-correlation (GOC) in resistant (RES) but not in susceptible (SUS) phenotypes.** Only genes with significant gain-of-correlations at FDR < 0.05 were analyzed using GOstats. Odds ratios (OR) for the GO enrichment of terms were listed. The top ten significantly enriched GO terms from each GO category are shown. The color of GO term bars denotes each of the three GO categories: Biological Process (BP), Cellular Component (CC), and Molecular Function (MF). N = 20 per group.

Gene-trait differential correlation analysis was also conducted using DGCA. Multiple genes were found to be strongly correlated with worm count data ($|\rho| > 0.65$). In resistant lambs, *DTX2* and *AQP3* were negatively and positively correlated with worm counts ($\rho = -0.77$; $P = 6.82 \times 10^{-5}$ and $\rho = 0.69$; $P = 0.0008$), respectively (S9 Table). Moreover, these two genes were differentially correlated with this trait between the two groups, that is, no correlation between them and worm counts in susceptible lambs. On the other hand, Kazal-type serine peptidase inhibitor domain 1 (*KAZALD1*) and Zinc Finger Protein 467 (*ZNF467*) were positively correlated with worm counts in SUS, $\rho = 0.80$ and 0.70, respectively ($P < 0.0006$). In addition, calponin 2 (*CNN2*) and TATA-box binding protein associated factor 13 (*TAF13*) were negatively correlated with worm counts in susceptible animals ($P < 0.0004$). These four genes, *CNN2, KAZALD1, TAF13, ZNF467*, did not appear to be correlated with worm counts in RES animals, though. In addition, several genes were also differentially correlated with another parasitological trait, WEC values obtained from the lambs grazing on pasture. For example, mitochondrial pyruvate carrier 1 (*MPC1* or *SLC54A1*), a key metabolic protein that regulates the transport of pyruvate into the mitochondrial inner membrane, is strongly yet negatively correlated with WEC values ($\rho = -0.84$; $P = 2.95 \times 10^{-6}$), but only in RES animals. All other genes with differential correlation with WEC can be found in S10 Table.

Using 255,753 correlated gene pairs at FDR < 0.05 as an input, the Planar Filtered Network (PFN) construction in the MEGENA pipeline detected 11,820 edges and resulted in a total of 195 modules based on the default multiscale clustering analysis thresholds. In these modules, there were 265 unique hub genes. For example, UDP-GlcNAc:betaGal beta-1,3-N-acetylglucosaminyltransferase 3 (*B3GNT3*) served as a hub gene for modules ME158 and ME391, while hexokinase 3 (*HK3*) acted as a hub gene for three modules, ME197, ME403 and ME460. The enrichment of GO terms in these modules were analyzed. Select GO BP terms in a subset of 14 modules are shown in Fig 6. Three BP terms, receptor transactivation, adenylate cyclase-activating adrenergic receptor signaling, and isocitrate metabolic process, were significantly enriched in ME309. Four BP terms, electron transfer activity, respiratory electron transport chain, ribose phosphate biosynthetic process, and oxidoreduction-driven active transmembrane transporter activity, were enriched in ME24. The BP leukocyte migration involved in inflammatory response was significantly enriched in three modules, ME197 (OR = 63.19; FDR = 7.88 x 10$^{-4}$), ME403 (OR = 114.77; FDR = 2.62 x 10$^{-4}$) and ME440 (OR = 204.25; FDR = 9.45 x 10$^{-5}$). Cell adhesion molecule binding and cadherin binding were also significantly enriched in the three modules, ME24, ME156, and

| GO Term | ME003 | ME010 | ME011 | ME024 | ME025 | ME156 | ME197 | ME297 | ME309 | ME335 | ME389 | ME403 | ME440 | ME460 | $-\log_{10} P$ |
|---|---|---|---|---|---|---|---|---|---|---|---|---|---|---|---|
| protein localization to membrane | 2.19 | | | | | | 4.67 | | | | | | 2.38 | | |
| protein-containing complex organization | 1.69 | | | 2.20 | | | | | | | | | 1.79 | | 6 |
| protein-containing complex assembly | 1.79 | | | 2.28 | | | | | | | | | 1.92 | | 5 |
| positive regulation of protein localization to membrane | 4.42 | | | | | | | | | | | | 4.36 | | 4 |
| positive regulation of biosynthetic process | | 2.30 | | | | | | | | 5.61 | | | | | 3 |
| appendage development | | 6.47 | | | | | | | | | | | | | 2 |
| limb development | | 6.47 | | | | | | | | | | | | | |
| embryonic limb morphogenesis | | 8.04 | | | | | | | | | | | | | |
| cytoplasmic translation | 2.99 | | 6.53 | | 2.72 | | | 5.74 | 3.31 | | | | 3.86 | | |
| structural constituent of ribosome | | | 6.14 | | | | | | | 2.23 | | | 2.59 | | |
| RNA binding | | | 2.64 | 1.89 | | | | | | | | | | | |
| oxidoreduction-driven transmembrane transporter activity | | | | 21.21 | | | | | | | | | | | |
| ribose phosphate biosynthetic process | | | | 8.85 | | | | | | | | | | | |
| respiratory electron transport chain | | | | 13.02 | | | | | | | | | | | |
| electron transfer activity | | | | 15.22 | | | | | | | | | | | |
| structural molecule activity | 1.75 | | 3.13 | | 4.21 | | | 4.54 | | | | | 1.98 | | |
| regulation of gene expression | | | | | | 1.94 | | | | 4.00 | 1.92 | | | | |
| cadherin binding | | | | 3.26 | | 3.45 | | | | | | 3.71 | | | |
| cell adhesion molecule binding | | | | 2.96 | | 3.14 | | | | | | 3.39 | | | |
| regulation of stress granule assembly | | | | | | 130.97 | | | | | 139.74 | | | | |
| positive regulation of organelle assembly | | | | | | | 22.83 | | | | | 42.69 | | 128.72 | |
| regulation of podosome assembly | | | | | | | 110.64 | | | | | 200.94 | | 91.91 | |
| podosome assembly | | | | | 18.85 | | 73.74 | | | | | 133.92 | | | |
| regulation of alkaline phosphatase activity | | | | | | | 442.76 | | | | | 189.18 | | 321.95 | |
| regulation of transcription by RNA polymerase II | | 1.82 | | | | | | | | 6.90 | | | | | |
| positive regulation of transcription by RNA polymerase II | | 2.49 | | | | | | | | 9.73 | | | | | |
| regulation of DNA-templated transcription | | | | | | | | | | 6.72 | | | | | |
| ER stress-induced intrinsic apoptotic signaling pathway | | | | | | | | | | 71.37 | | | | | |
| substrate-dependent cell migration, cell extension | | | | | | 65.47 | | | | | 69.86 | | | | |
| leukocyte migration involved in inflammatory response | | | | | | | 63.19 | | | | | 114.77 | 204.25 | | |
| cellular defense response | | | | | | | 27.61 | | | | | 50.14 | | 89.24 | |
| cellular nitrogen compound metabolic process | | | | | | | | | | | | | | 14.33 | |
| cellular amide metabolic process | | | | | | | | | | | | | | 9.31 | |
| receptor transactivation | | | | | | | | | >500.00 | | | | | | |
| adenylate cyclase-activating adrenergic receptor signaling | | | | | | | | | >500.00 | | | | | | |
| isocitrate metabolic process | | | | | | | | | 37.30 | | | | | | |
| positive regulation of phosphatidylinositol 3-kinase signaling | | | | | | 20.74 | | | | | | 38.79 | | 41.88 | |

**Fig 6. Significantly enriched Gene ontology (GO) terms in select modules detected using the DGCA algorithm.** Top: each column represents a module (ME). Left: significantly enriched GO terms at FDR adjusted *P*-value < 0.01 in select modules. Right: the color panel represents significance threshold (-log$_{10}$ *P* value). The color in each cell denotes significance levels. The number in each cell represents odds ratio (OD).

ME389. Intriguingly, ME11 represents a differential correlation module with most enriched gene pairs with GoC in RES lambs (Fig 7), whereas ME30 contained the most enriched gene pairs with GOC in SUS animals (S4 Fig). ME11 is rather large with 125 genes (nodes) and 669 edges (connections). The gene pairs with RES-specific GOC can be found in S11 Table. Three genes, olfactory receptor family 2 subfamily AT member 4 (*OR2AT4*), RAB32, member RAS oncogene family (*RAB32*, a gene encoding a Rab GTPase that acts as critical regulator of a host defense pathway leading to eliminate bacterial pathogens) [37], and TNF receptor superfamily member 1A (*TNFRSF1A*), served as hubs in this module with 104, 50, and 74 edges, respectively. Moreover, *OR2AT4* is positively correlated with CD163 only in RES animals. Of note, several genes, such as *KRT8* and *LGALS4*, were also important members in this module. *KRT8* had 14 edge connections in this module; and 8 of them were positively correlated with other genes in both RES and SUS animals. For example, *KRT8* was significantly correlated with NAD kinase (*NADK*) and valosin containing protein (*VCP*) in both RES and SUS animals. Four of the nine correlations between *LGALS4* and other genes, such as *VCP* and cadherin related family member 2 (*CDHR2*), were also positive in both RES and SUS animals. Nevertheless, *KRT8* was differentially correlated

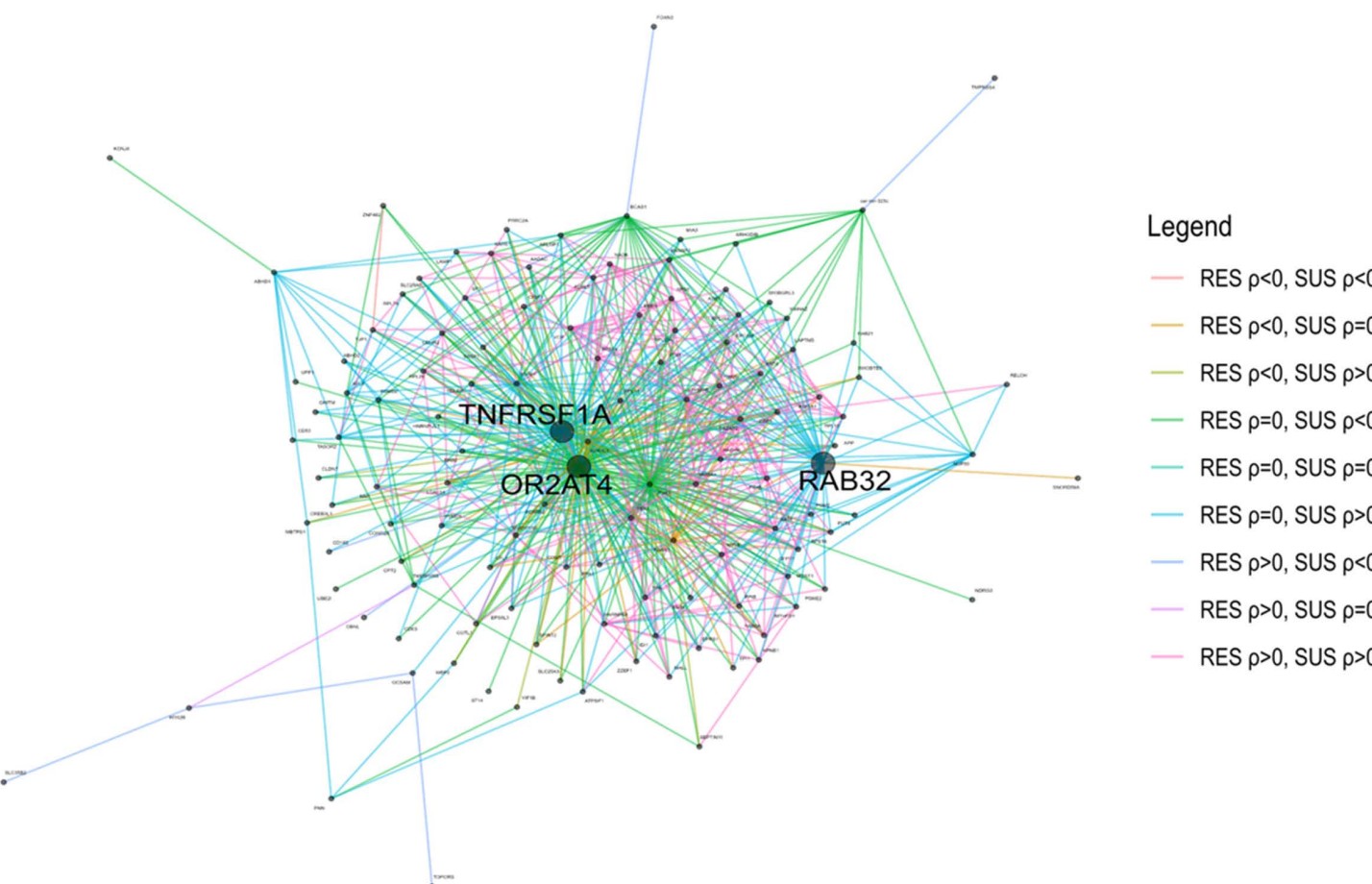

**Fig 7. A differential correlation module from DGCA-MEGENA analysis with most enriched in gene pairs with gain-of-correlation (GOC) in RES lambs.** The sizes of nodes and texts of gene symbols are proportional to the number of connections (correlations) for each gene. Edges are colored according to the differential correlation class as described in the legend (Right), while edge weight is proportional to the absolute value of the z-score for the difference in correlations between the two phenotypes. Three hub genes, olfactory receptor family 2 subfamily AT member 4 (*OR2AT4*), RAB32, member RAS oncogene family (*RAB32*), and TNF receptor superfamily member 1A (*TNFRSF1A*), each correlated with 95, 19, and 16 other genes, respectively, were shown in a large bold font. $\rho$: Spearman rho correlation coefficient. The plot was made using Cytoscape (v3.10.3).

with two of the three hub genes, *RAB32* and *TNFRSF1A*, only in SUS lambs and had no correlations with these two hubs in RES animals. Intriguingly, all correlations between a microRNA, oar-mir-323c, and 13 other genes, including Rho GDP dissociation inhibitor beta (*ARHGDIB*), Rho related BTB domain containing 3 (*RHOBTB3*), and lysosomal protein transmembrane 5 (*LAPTM5*), were negative in SUS animals.

## Discussion

Over past decades, numerous efforts have been made to identify genetic variants associated with parasite resistance in sheep; and as many as 126 significantly associated markers and genomic regions have been implicated in WEC and related parasitological traits for major sheep parasites, such as *H. contortus* and *T. circumcincta* [38]. However, no chromosomal regions attained genome-wide significance for quantitative trait loci (QTL) influencing resistance to *T. colubriformis* (WEC) were detected [39]. Only one region attained chromosome-wide significance, and five other regions achieved point-wise significance for the presence of QTL affecting WEC. For example, QTL on chromosomes 6 and 3 had a logarithm of the odds (LOD) score of 4.2 and 3.9 for two WEC measurements, obtained after each of the two rounds of 20,000 *T. colubriformis* L3 challenge infections, respectively. The results from this microsatellite-based analysis did not provide convincing evidence for the presence of major genes affecting *T. colubriformis* resistance in sheep. While several recent studies have utilized genome-wide association analyses (GWAS) with thousands of single nucleotide polymorphism (SNP) markers to identify genomic regions associated with parasite resistance in sheep, none of them specifically investigated resistance to *T. colubriformis* [40–43]. As a result, genes and biological pathways underlying host resistance to *T. colubriformis* infections in sheep remain largely unknown.

Expressed resistance to GI nematode infections in sheep includes decreased worm establishment, arrested development of invading larvae, reduced parasite fecundity, and enhanced worm expulsion. Early studies using an elegant system compared immune responses of immune sheep established via three rounds of truncated *T. colubriformis* infections with those of parasite naïve sheep of the same age in response to a *T. colubriformis* challenge infection [44]. Compared with naive sheep, immune animals, which had a 94% reduction in worm burden, displayed a significantly increased level of parasite-specific IgG$_1$ and IgA, and histamine levels, as well as more globule leucocytes in the lamina propria. Moreover, the mucus of immune lambs possessed anti-larval activities that resulted in the displacement of larvae from their predilection site two to three hours post challenge infection [44]. An early microarray analysis demonstrated that a global down-regulation of many genes involved in immune functions, including antigen presentation, caveolar-mediated endocytosis and protein ubiquitination in intestinal lymph nodes of lambs in response to a challenge infection with *T. colubriformis* [17]. In this study, we compared two groups of sheep with contrasting differences in resistance and susceptibility to parasitic infections in response to challenge infections. The resistant lambs in our study had a significant reduction in worm establishment, as evidenced by a 98.8% reduction in worm burden after challenge infection, similar to what was observed in the immune lambs used in published studies. Unlike those immune lambs established by truncated infections, the resistant phenotype in our lambs were stable and well differentiated and likely under moderate genetic control.

Three genes encoding galectins, *LGSLA15*, *LGALS3* and *LGALS4*, displayed significant differences in relative abundance between the RES and SUS groups in our study. *LGALS3* and *LGALS4* had significantly higher transcript abundance in SUS, whereas the mRNA level of *LGSLA15* was five times higher in RES than in SUS. *LGALS3* is known to be regulated during parasitic infections in various host-parasite systems [45], such as the mouse-*Toxoplasma gondii* system [46]. While the expression of *LGALS3* in ovine whole blood can be upregulated by garlic supplementation [47], its involvement in GI nematode infections in small ruminants has not been reported before. *LGSLA15* has been known to play a critical role in parasite immunity in sheep [48], however the role of *LGALS4* in host immune responses in sheep remains unknown. The galectin encoded by *LGSLA15*, also known as ruminant specific galectin-11, is produced by epithelial cells in the GI tract and then secreted into the mucous lining [49]. LGSLA15 protein binds to carbohydrate moieties on the surface of the L4 and adult stages of the parasite but not to the exsheathed L3 stage of *H. contortus in vitro.* Recombinant

LGALS15 displays a potent antiparasitic activity and inhibits larval growth and development in an isoform-dependent manner [49]. While incubated with the homotetrameric LGALS15 recombinant protein (isoform 1), encoded by natural genetic variant 1, 85% of exsheathed L3 stage of *H. contortus* were unable to develop to L4 larvae. In contrast, those exposed to monomeric–dimeric isoform 2 developed normally [50], suggesting that oligomerization status of LGALS15 is critical for its antiparasitic activity. It is conceivable that the frequency and distribution of the genetic variants in *LGALS15* likely represent one of the key factors in determining host resistance and susceptibility to GI nematode infection in sheep. Nevertheless, no QTL or genomic loci spanning the *LGALS15* gene have so far been implicated in parasite resistance. Our future work will focus on the identification of genetic variants in both coding and promoter regions of this gene in natural sheep populations and different breeds to understand its transcription regulation and splicing mechanisms.

The majority of the DEG detected in our study had significantly higher abundance in SUS than in RES lambs. Multiple pathways involved in host immunity were significantly enriched among these genes, notably, antigen processing and presentation and Fc gamma R-mediated phagocytosis. It is well known that these pathways play an essential role in host defense mechanisms, particularly against bacterial pathogens. Their roles in mounting a strong host immune response to *T. colubriformis* infection remain to be unraveled, however it is clear that many of the processes upregulated in SUS group animals were not effective in removing parasite infections since these animals had high worm burdens at necropsy. Coincidentally, five genes encoding variable (V) regions of T cell receptors had significantly higher expression in RES lambs in response to *T. colubriformis* infection. It would be interesting to establish causal relations between these genes and antigen presentation in sheep, especially as these are associated with animals that mounted an effective response, eliminating parasites prior to necropsy.

Various algorithms and pipelines developed to analyze differential gene expressions, such as edgeR and DESeq2, rely on statistical modeling to identify genes with substantial differences in expression levels (usually ≥1.5 to 2.0 fold) between conditions or disease status. While these algorithms are powerful in rapidly analyzing count data from thousands of genes obtained using RNAseq, their limitations are also obvious, such as false positives (negatives), batch effects, and the inability to reliably detect subtle changes in transcript abundance between conditions. For example, subtle changes in the expression level of transcription factors can have important biological consequences and lead to variations in phenotypes. To overcome these limitations, we examined global gene correlation patterns between resistance and susceptibility to infection in the present study as a complement to differential gene expression analyses. Gene correlation analysis relies on statistical methods, such as Pearson's correlation, Spearman rank, and/or biweight midcorrelation, to quantify relationships between genes, aiming to identify genes that may function together or be regulated in a coordinated manner. These analyses can provide unique insights into biological processes implicated in the development of host resistance and resilience to parasitic infections. In this study, we used two different statistical methods for gene correlation analyses, biweight midcorrelation or bicor [51] as implemented in R as part of WGCNA and Spearman's rank in DGCA.

The significant gene pair correlations identified were more numerous in susceptible lambs than in resistant lambs in our study. There were 4.65 times more significantly correlated gene pairs that were unique to SUS than those to RES. The total number of significantly correlated gene pairs identified using DGCA followed a similar trend, 3.41 times more in SUS than in RES. Moreover, WGCNA detected 53 times more modules in SUS than in RES, suggesting that the correlated genes in SUS likely share similar regulatory mechanisms and can be functionally related in the context of biological pathways. Intriguingly, five of the 153 modules detected using WGCNA in SUS were significantly correlated with parasitological traits. For example, MElightcran1 was positively correlated with worm counts ($R = 0.67$; $P = 0.001$), whereas MEtan3 and MEpink3 were positively correlated with both WEC and worm burden ($P < 0.05$). MElightcran1 consisted of 96 highly interacted or correlated genes; but only one of them was differentially expressed; as a result, conventional DEG analyses alone would have missed the majority of these genes. 50 of the 96 genes in this module are protein-coding while the remaining were non-coding, including 29 lncRNA and one microRNA (oar-mir-221). One unannotated gene acted as a hub

gene. Selective breeding had somehow disrupted the gene interactions within this module; and no corresponding modules were preserved in the resistant lambs.

PIGR transports multimeric IgA and IgM across intestinal epithelia [52]; and IgA plays a critical role in host-parasite interactions, shaping sheep-*T. colubriformis* relations [13]. The clearance of a protozoan parasite, *Giardia muris*, is severely compromised in *PIGR*-deficient mice [53]. Strong *PIGR* mRNA expression has been implicated in the development of host resistance to parasite infections in cattle [54]. Our present data show that in WGCNA networks, *PIGR* was one of the key members in MEturquoise with module membership $k_{ME}$ = 0.99 ($P$ = 3.46 x 10$^{-17}$), whereas its *GS* value for the WEC trait is negative ($GS$ = -0.42). In DGCA-MEGENA networks, *PIGR* was negatively correlated with 30 genes in SUS but not correlated with these genes in RES lambs. Together, our data show that *PIGR* expression is negatively correlated with two parasitological traits, WEC and worm burden. Nevertheless, its expression was not differentially expressed between the RES and SUS phenotypes, as defined by our cutoff thresholds (FDR = 0.1021). This piece of data provided further support that the methods for differential expression analysis and gene correlation analysis complement each other. Moreover, we hypothesize that dietary approaches to disrupt close gene interactions within MElightcran1 or other key modules may have potential to reduce worm burden. If this proves feasible, we could screen plant forages, particularly those tannin-rich forage, phytochemicals or feed additives that specifically disintegrate gene interactions within certain modules for sustainable parasite controls. Indeed, as reviewed in a recent article [55], at least 59 published trials have been conducted to investigate antiparasitic activities of tannin-rich plants. The majority of these trials have shown a varying efficacy of these plants in reducing WEC or worm burden [55]. For example, a long-term ingestion of sericea lespedeza forage rich in condensed tannins (80 g/kg Dry Matter or DM) by Merino sheep for 35 days resulted in a significant reduction in WEC ($P$ = 0.05), compared to a control diet with low tannin contents (< 1.50 g/kg DM) [56]. These studies have demonstrated the feasibility of forage based approaches for parasite controls.

DGCA also identified four genes with strong (positively and negatively) correlations with worm counts in SUS but not in RES animals (FDR < 0.001). *KAZALD1* and *ZNF467* (as well as *ZNF831*) were positively correlated with worm counts only in SUS, whereas *CNN2* and *TAF13* were negatively correlated with worm counts. As a key member of the calponin family, *CNN2* is known to be involved in smooth muscle contractility and cell migration [57]. It also affects cell proliferation and apoptosis [58]. Accelerated smooth muscle contractility contributes to worm expulsion [59,60]. The expression of *CNN2* can be regulated via various methods [61,62]. Because the correlation between *CNN2* and worm burden was negative, it would be interesting to test whether increased *CNN2* expression using experimental approaches could reduce worm burden *in vivo*. Conversely, both ZNF467 and ZNF831 act as transcription factors in model organisms. The former is involved in histone deacetylase complexes while the latter acts as a transcriptional repressor targeting the STAT3/Bcl2 signaling pathway [63]. *KAZALD1* is involved in colon tumor development [64] and plays a role in wool fitness in sheep [65]. However, these genes have not been implicated in host-parasite interactions. While gene correlation analyses provided new insights into the mechanisms underlying host resistance and susceptibility to parasite infections, more experimental evidence is needed to establish causal relationships between these genes and parasitological phenotypes.

In conclusion, selective breeding for parasite resistance significantly reduced worm establishment in sheep in response to a *T. colubriformis* challenge infection. The manifestation of resistance or susceptibility becomes evident in the host transcriptome at the site of infection as demonstrated by alterations in transcript abundance of 319 genes as well as 34 enriched pathways involving in a broad array of biological processes and molecular function, including those related to host immunity, basic metabolism, and signaling. The genes in susceptible animals tend to be more closely correlated and organized in discrete clusters or modules. Multiple modules were positively correlated with worm counts and WEC, but only in susceptible lambs. Many correlated gene pairs unique to each of the two phenotypes have the potential to be developed as biomarkers. Developing experimental approaches to disrupt gene interactions in key modules may represent a novel strategy for sustainable parasite control in small ruminants.

## Supporting information

**S1 Fig. Principal Component Analysis (PCA) plot showing the overall transcriptome structure in sheep.** The plot was generated using top 15,000 genes sorted by their median values in transcript abundance. RES: Resistant lambs; SUS: Susceptible lambs. $N=20$ per group.
(TIF)

**S2 Fig. Module trait relationships in the gene interaction network inferred from parasite resistant lambs.** Left panel: Modules represented colors. Right: Correlation coefficient calculated using the biweight mid-correlation (bicor) method. The numbers in each cell represent correlation coefficient and $P$ value (in the parenthesis). The blue module (MEblue) had a negative correlation with fecal worm egg counts (WEC) R = -0.53 ($P=0.02$). $N=20$ per group.
(TIF)

**S3 Fig. Differentially correlated gene pairs detected using the Differential Gene Correlation Analysis (DGCA) algorithm.** Leucine rich repeat containing G protein-coupled receptor 4 (LGR4); cysteine-serine-rich nuclear protein 3 (CSRNP3). RES: Resistant lambs. SUS: Susceptible lambs. $N=20$ per group.
(TIF)

**S4 Fig. The differential correlation module (ME30) most enriched in genes with gain of correlation (GOC) in parasite susceptible (SUS) lambs but no significant correlations in resistant (RES) lambs.** Node size and gene symbol text size are proportional to the number of correlations or connections for each gene. Edges are colored according to the differential correlation class (see Legend), while edge weight is proportional to the absolute value of the z-score for the difference of correlation between RES and SUS lambs. USE1 (Unconventional SNARE in the ER 1) acted as the hub gene in this module. $\rho$: Spearman rho correlation coefficient.
(TIF)

**S1 Table. The sample metadata with parasitological traits.**
(XLSX)

**S2 Table. Quantitative RT-PCR primers in sheep.**
(XLSX)

**S3 Table. The basic statistics of raw sequences in all 40 samples.**
(XLSX)

**S4 Table. Differentially expressed genes (DEG) detected using DESeq2.** 319 genes with significantly different transcript abundance between resistant (RES) and susceptible (SUS) lambs in response to Trichostrongylus colubriformis challenge infection at a combined cutoff FDR < 0.05 and |fold change| > 1.50. $N=20$ per group.
(XLSX)

**S5 Table. Significantly enriched keywords in the UniProt Knowledgebase (UniProtKB) among the genes with significantly higher abundance in susceptible lambs.**
(XLSX)

**S6 Table. Select Kyoto Encyclopedia of Genes and Genomes (KEGG) pathways enriched among genes with significantly higher transcript abundance in susceptible lambs.**
(XLSX)

**S7 Table. Kyoto Encyclopedia of Genes and Genomes (KEGG) pathways significantly enriched in Module turquoise in parasite resistant lambs.**
(XLSX)

**S8 Table. Gene Ontology (GO) analysis among the correlated gene pairs unique to either resistant (RES) or susceptible (SUS) lambs.** FDR: false discovery rate.
(XLSX)

**S9 Table. Genes significantly correlated with worm counts.** RES: parasite resistant; SUS: parasite susceptible. cor: correlation coefficient. pVal: adjusted *P* value.
(XLSX)

**S10 Table. Genes significantly correlated with fecal worm egg counts (WEC).** RES: parasite resistant; SUS: parasite susceptible. cor: correlation coefficient. pVal: adjusted *P* value.
(XLSX)

**S11 Table. The edges (connections) unique to resistant lambs (RES) in the differential.**
(XLSX)

## Acknowledgments

The authors thank Oak Ridge Institute for Science and Education (ORISE) for sponsorship of FL during her postdoctoral training at USDA-ARS. Additional technical support and animal care was provided by CSIRO staff Troy Kalinowski, Dan Driscoll, Dominic Niemeyer and Graham Acton. Mention of trade names or commercial products in this publication is solely for the purpose of providing specific information and does not imply recommendation or endorsement by the US Department of Agriculture (USDA).

## Author contributions

**Conceptualization:** Peter W Hunt, Robert W. Li.

**Data curation:** Fang Liu.

**Formal analysis:** Fang Liu, Jonathan Shao.

**Funding acquisition:** Robert W. Li.

**Investigation:** Fang Liu, Jody McNally, Aaron B. Ingham, Peter W Hunt, Robert W. Li.

**Methodology:** Fang Liu.

**Project administration:** Robert W. Li.

**Software:** Fang Liu.

**Supervision:** Peter W Hunt, Robert W. Li.

**Visualization:** Fang Liu.

**Writing – original draft:** Fang Liu, Robert W. Li.

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
