## [Decision Letter · Decision Letter 0]

25 Jun 2025

Disruptions in gene interaction networks abolish host susceptibility to Trichostrongylus colubriformis infections in sheep

Dear Dr. Li,

Thank you for submitting your manuscript to PLOS Neglected Tropical Diseases. After careful consideration, we feel that it has merit but does not fully meet PLOS Neglected Tropical Diseases's publication criteria as it currently stands. Therefore, we invite you to submit a revised version of the manuscript that addresses the points raised during the review process.

Please submit your revised manuscript within 60 days Aug 24 2025 11:59PM. If you will need more time than this to complete your revisions, please reply to this message or contact the journal office at plosntds@plos.org. Please include the following items when submitting your revised manuscript:

We look forward to receiving your revised manuscript.

Kind regards,

Feng Xue, Ph.D.

Guest Editor

Jong-Yil Chai

Section Editor

Shaden Kamhawi

co-Editor-in-Chief

Paul Brindley

co-Editor-in-Chief

**Journal Requirements:**

**Reviewers' Comments:**

Reviewer's Responses to Questions

**Key Review Criteria Required for Acceptance?**

**Methods**

-Are the objectives of the study clearly articulated with a clear testable hypothesis stated?

-Is the study design appropriate to address the stated objectives?

-Is the population clearly described and appropriate for the hypothesis being tested?

-Is the sample size sufficient to ensure adequate power to address the hypothesis being tested?

-Were correct statistical analysis used to support conclusions?

-Are there concerns about ethical or regulatory requirements being met?

Reviewer #1: Introduction:

Please consider adding a brief taxonomic description of T. colubriformis (e.g., family Trichostrongylidae, morphology, site of infection). Also, the zoonotic relevance, mentioned in the author summary, could be more clearly emphasized in the main text. In support of this, we suggest including a few additional references specific to Trichostrongylus biology and pathogenesis (e.g., larval development stages, immune evasion mechanisms, host pathology).

Line 81: When referring to the zoonotic potential of T. colubriformis and T. axei, consider citing human case reports or prevalence data, especially from tropical regions.

Parasitological Methodology:

Some moderate adjustments are recommended for clarity:

- How were worms counted? Was the entire small intestine examined, or was a subsampling protocol used?

- Were sex and/or developmental stages of the worms recorded? This could offer insights into fecundity or developmental arrest.

Fecal Egg Count (FEC):

Please clarify the following:

- Were all animals at 0 EPG prior to infection?

- Was there any egg shedding observed post-infection in the resistant group?

- If possible, more explicitly correlate FEC data with worm counts in the Results section.

Reviewer #2: See Summary and General comments

Reviewer #3: Line 136 - Haemonchus selection and Trichostrongylus selection flocks – same flock or two flocks?

Line 157 – why was a sub sample of 30ml taken? Would this have captured the number of worms appropriately particularly after freezing the sample?

Reviewer #4: (No Response)

**Results**

-Does the analysis presented match the analysis plan?

-Are the results clearly and completely presented?

-Are the figures (Tables, Images) of sufficient quality for clarity?

Reviewer #1: - Host–Parasite Interaction:

A brief discussion linking the identified immune pathways with known mechanisms of parasite immune evasion or modulation by T. colubriformis would strengthen the parasitological interpretation of the findings.

Specific Points in the Text:

- Line 215: There appears to be a repetition: “resistant lambs lambs differed markedly...”. Suggested rephrasing:

“Worm establishment was significantly impaired in resistant lambs.”

Figures:

- The current figures are clear. However, we suggest including images of the parasite (if available), to visually reinforce the target organism and improve accessibility for readers less familiar with T. colubriformis.

Reviewer #2: See Summary and General comments

Reviewer #3: Overall the result are difficult to follow. Several of the paragraphs encompassing whole pages need breaking down into smaller paragraphs with distinct thoughts/ideas. Equally including long lists of genes etc is sometimes difficult to read. The rationale/context of the results was sometimes unclear for someone with not a high level of experience in bioinformatics - some of this was better explained in the discussion, which could be moved to the results section.

Line 215 – this section should start with the parasite data before discussing the transcriptome results.

Line 223 – only 7 DEG in resistant lambs – is this reflective of the sample location?

Line 260 – needs an explanation as to why gene interaction networks used? Also while I appreciate it is standard practice to refer t the modules as colours but is there another method that can be used that is more informative/easier to read.

Reviewer #4: (No Response)

**Conclusions**

-Are the conclusions supported by the data presented?

-Are the limitations of analysis clearly described?

-Do the authors discuss how these data can be helpful to advance our understanding of the topic under study?

-Is public health relevance addressed?

Reviewer #1: The mention of feed/phytochemical strategies to modulate gene networks is innovative and promising. Consider citing examples of plant-based anthelmintics already used in ruminants (e.g., tannin-rich forages like Lespedeza) to contextualize this idea.

- Lines 604–610: A table or figure summarizing key genes and their associated phenotypes (e.g., DEGs correlated with worm burden) would enhance readability for parasitologists.

Reviewer #2: See Summary and General comments

Reviewer #3: (No Response)

Reviewer #4: (No Response)

**Editorial and Data Presentation Modifications?**

Reviewer #1: (No Response)

Reviewer #2: See Summary and General comments

Reviewer #3: (No Response)

Reviewer #4: (No Response)

**Summary and General Comments**

Reviewer #1: The manuscript “Disruptions in gene interaction networks abolish host susceptibility to Trichostrongylus colubriformis infections in sheep” is comprehensive, well-structured, and presents a high-quality investigation into the molecular basis of resistance to this nematode in sheep. The experimental design is sound, the sample size appropriate, and the analyses appear thoroughly applied and well interpreted. The study is a valuable contribution to the field of parasitology, particularly in host–parasite interaction research.

The manuscript clearly demonstrates phenotype-based differences in helminth burden between resistant and susceptible animals. The integration of parasitological data with transcriptomic analysis is commendable, especially the strong reduction in worm burden (98.8%), which supports the study's conclusions. From a practical standpoint, the findings offer potential genetic markers for selective breeding and lay the groundwork for future anti-parasitic strategies.

Reviewer #2: This large transcriptional dataset has an interesting experimental design and represents valuable host-response data to infection. Having 20 replicates enabled the correlation analysis, which wouldn't be possible with a smaller sample size spanning a smaller range of worm burden values. However, several improvements to the analysis are necessary to improve clarity and transparency of the data:

- Importantly, a thorough overview of the dataset should be provided. This should include a supplementary table that includes, for each sample, worm counts, egg counts and any other relevant metadata, the total number of reads sequenced, the total number of reads mapped, and the matching accessions on the SRA upload, such that the analysis could be repeated by other researchers if they wanted to. The distribution of read counts is important for interpreting the data, since samples with very low coverage will result in more genes being below detection level, and subsequently they will lack the ability to have correlation with other genes. Hopefully there is no substantial difference in these metrics between RES and SUS.

- At minimum, a PCA plot using gene expression across all genes would also show how well the samples correlate with each other, and how much the RES and SUS separate. Additionally, for example, the first principal component could be plotted against total worm counts, to show whether the overall gene expression pattern correlates with infection burden.

The PCA is important because of the correlation analysis. It’s stated that “DGCA detected approximately four times more unique gene correlation pairs in SUS than in RES lambs.” This implies that RES lambs have more noisy overall expression patterns, which should show us as them being scattered on the clustering.

Another reason that SUS may have more correlated gene pairs is that the SUS animals are undergoing substantially more of an immune response, so many sets of genes are being concurrently upregulated across animals. If the RES lambs are not responding as strongly, then they would be closer to their usual baseline uninfected transcriptional phenotype.

- It would also be helpful to have a large Excel supplementary table that includes at least the read counts and FPKM values for each gene in each sample, so that readers wouldn’t need to reprocess the raw data to check the expression level across samples for a gene of interest.

- Authors may consider using Ensembl biomart to bulk-download functional annotations for their genes from a variety of functional annotation algorithms such as Interpro and panther, which may provide annotations for many genes that currently lack it in the supp tables.

- Some or most of the first paragraph and a half of the discussion (up to line 485) would be more appropriate to be in the introduction.

- Line 536, batch effects can be considered in DESeq by incorporating them as covariates in the model. False positives and negatives are likely to be just as bad or worse in a correlation analysis, and “subtle” but significant changes can be captured easily by DESeq simply by omitting a fold change cutoff and only considering the P value, which is acceptable practice. The main reason to use correlation instead of DESeq for your analysis is that you want to look at gene changes over a continuous variable and not perform a simple pairwise comparison. That’s an acceptable reason that’s more appropriate to mention than the ones in this sentence.

For example, “ME lightcran1 consisted of 96 highly interacted or correlated genes; but only one of them was differentially expressed; as a result, conventional DEG analyses alone would have missed the majority of these genes.” How many of these genes had significant FDR values but just lacked the sufficient fold change cutoff?

- The Figure 1 caption should indicate the statistical test used (I’m assuming it’s the Wilcoxon rank sum test based on the methods). Was a test for normality ran for the control group for each data type to test whether the data follows a normal distribution? If it does, then a T-test is more appropriate and will likely yield better P values.

- Figure 5 caption says that it shows GO terms with GOC in RES but not SUS, but line 358 says that it’s showing ones with similar patterns between RES and SUS, which matches what the bar graphs are showing. So, it’s unclear what’s shown here, and the odds ratio doesn’t indicate significance, which would be the more useful metric (maybe the -Log of the P value). It would be interesting to split this into the top 10 shared terms, the top 10 RES-only terms and the top 10 SUS terms rather than the top 30 shared.

- Figure 5 caption should indicate which color goes with which GO parent term.

- Figure 6 isn’t useful without short GO term descriptions to understand what the data is showing, since readers can’t be expected to know or manually search each GO ID. The column headers could just have the number (“003” instead of “ME003”) and could have only one decimal place in the table, to make them more narrow.

- Some of the correlation network analysis is confusing. For example, Figure 7 contains the most enriched gene pairs with GoC in RES, but if you look at the hub gene RAB32, almost all of its edges are blue, which the legend indicates means there is a positive correlation in SUS but not RES. It’s difficult to see in the other two, but TNFRSF1A also appears to have mostly blue edges, and OR2AT4 maybe green which means negative in SUS and nothing in RES. Overall it’s not obvious from the figure how this network is enriched for RES over SUS, where we’d expect to see more purple and orange and less blue and green.

However, I understand that there is more overall correlation in SUS compared to RES, so most networks would look like this. It does look like some genes, most notably KANK1, SIPA1L1, have RES-specific edges. Because of this, it would make more sense to focus on the nodes like this with the most RES-specific edges, and perhaps have a second panel to Figure 7 which only shows the edges that are RES-specific or opposite direction between RES and SENS, to highlight the informative parts of this network.

Reviewer #3: The manuscript by Liu and colleagues describes the genes/gene networks involved in host susceptibility to Trichostrongylus colubriformis infections in sheep. This is a valid study and the approach taken by the authors is very robust, particularly the use of sheep specifically bred for resistance/susceptibility to parasite infection. However, the manuscript in written in a format and uses terminology more suitable for a bioinformatics journal. To be published in PLoS NTD the authors should revise the manuscript to make it clearer why the different techniques were used and what the results mean. The discussion does do this in part, and several sections could be moved around to provide better context, but the authors need to think of the readership or consider a more bioinformatics journal. As detailed above the results section includes several long paragraphs that are difficult to follow, so should be edited into smaller paragraphs.

Some final minor points:

Abstract - should mention the sample used for transcriptomics to clarify that a GWAS study wasn't carried out.

Line 440 – Merino sheep rather than Merinoland?

Reviewer #4: The manuscript by Liu et al., describes the research on differentially expressed genes between susceptible and resistant sheep strain to the Trichostrongylus colubriformis infection. The research topic is important and interesting, but in its current state the manuscript needs thorough revision.

1. The list of Supplementary materials in the manuscript is missing, so it is unclear what the authors have provided as available materials and what they have not

2. There are no basic sequencing statistics, in particular, the average number of reads per library, quality, the percentage of aligned reads per genome, etc. Please provide

3. Please clearly indicate how many biological and technical replicates were used in preparing the DNA libraries, how many samples were there in total for sequencing. Did you take material from one animal for each library or pool them? This information should be in the M&M section

4. Please provide a PCA plot to demonstrate how different your samples were and what percentage of differences (variance) main components explain.

5. Please indicate the versions of all bioinformatics packages that you used throughout the text

6. Lines 194-195. The authors state that they did apply filtering before the Differential Gene Correlation analysis. However, it is unclear whether the authors filtered at the very beginning before identifying DEG genes? This is important and may affect the total number of DEGs. Please state clearly.

PLOS authors have the option to publish the peer review history of their article (what does this mean? ). If published, this will include your full peer review and any attached files.

**Do you want your identity to be public for this peer review?** For information about this choice, including consent withdrawal, please see our Privacy Policy .

Reviewer #1: **Yes: ** Eduardo Lopes-Torres

Reviewer #2: No

Reviewer #3: No

Reviewer #4: No

**Figure resubmission:**

**Reproducibility:**



---

## [Decision Letter · Decision Letter 1]

27 Jul 2025

Dear Dr. Li,

We are pleased to inform you that your manuscript 'Disruptions in gene interaction networks abolish host susceptibility to Trichostrongylus colubriformis infections in sheep' has been provisionally accepted for publication in PLOS Neglected Tropical Diseases.

Best regards,

Feng Xue, Ph.D.

Guest Editor

Jong-Yil Chai

Section Editor

Shaden Kamhawi

co-Editor-in-Chief

Paul Brindley

co-Editor-in-Chief

Reviewer's Responses to Questions

**Key Review Criteria Required for Acceptance?**

**Methods**

-Are the objectives of the study clearly articulated with a clear testable hypothesis stated?

-Is the study design appropriate to address the stated objectives?

-Is the population clearly described and appropriate for the hypothesis being tested?

-Is the sample size sufficient to ensure adequate power to address the hypothesis being tested?

-Were correct statistical analysis used to support conclusions?

-Are there concerns about ethical or regulatory requirements being met?

Reviewer #1: (No Response)

Reviewer #2: (No Response)

Reviewer #3: (No Response)

Reviewer #4: (No Response)

**Results**

-Does the analysis presented match the analysis plan?

-Are the results clearly and completely presented?

-Are the figures (Tables, Images) of sufficient quality for clarity?

Reviewer #1: (No Response)

Reviewer #2: (No Response)

Reviewer #3: (No Response)

Reviewer #4: (No Response)

**Conclusions**

-Are the conclusions supported by the data presented?

-Are the limitations of analysis clearly described?

-Do the authors discuss how these data can be helpful to advance our understanding of the topic under study?

-Is public health relevance addressed?

Reviewer #1: (No Response)

Reviewer #2: (No Response)

Reviewer #3: (No Response)

Reviewer #4: (No Response)

**Editorial and Data Presentation Modifications?**

Reviewer #1: (No Response)

Reviewer #2: (No Response)

Reviewer #3: (No Response)

Reviewer #4: (No Response)

**Summary and General Comments**

Reviewer #1: (No Response)

Reviewer #2: The authors have sufficiently addressed reviewer concerns

Reviewer #3: The authors have addressed my comments raised in the previous review.

Reviewer #4: (No Response)

PLOS authors have the option to publish the peer review history of their article (what does this mean? ). If published, this will include your full peer review and any attached files.

**Do you want your identity to be public for this peer review?** For information about this choice, including consent withdrawal, please see our Privacy Policy .

Reviewer #1: **Yes: ** Eduardo José Lopes-Torres or Lopes-Torres EJ

Reviewer #2: No

Reviewer #3: No

Reviewer #4: No

---

## [Editor Report · Acceptance letter]

Dear Dr. Li,

We are delighted to inform you that your manuscript, " 

Disruptions in gene interaction networks abolish host susceptibility to *Trichostrongylus colubriformis* infections in sheep," has been formally accepted for publication in PLOS Neglected Tropical Diseases.

Best regards,

Shaden Kamhawi

co-Editor-in-Chief

Paul Brindley

co-Editor-in-Chief
